# Orally Induced Hyperthyroidism Regulates Hypothalamic AMP-Activated Protein Kinase

**DOI:** 10.3390/nu13124204

**Published:** 2021-11-24

**Authors:** Valentina Capelli, Carmen Grijota-Martínez, Nathalia R. V. Dragano, Eval Rial-Pensado, Johan Fernø, Rubén Nogueiras, Jens Mittag, Carlos Diéguez, Miguel López

**Affiliations:** 1Department of Physiology, CIMUS, University of Santiago de Compostela-Instituto de Investigación Sanitaria, 15782 Santiago de Compostela, Spain; valentina.capelli@usc.es (V.C.); nathalia.romanelli@usc.es (N.R.V.D.); eva.pensado@usc.es (E.R.-P.); ruben.nogueiras@usc.es (R.N.); carlos.dieguez@usc.es (C.D.); 2CIBER Fisiopatología de la Obesidad y Nutrición (CIBERobn), 15706 Madrid, Spain; 3Unit of Internal Medicine and Endocrinology, Istituti Clinici Scientifici Maugeri, Department of Internal Medicine and Therapeutics, University of Pavia, 27100 Pavia, Italy; 4Department of Cell Biology, Faculty of Biology, Complutense University, 28040 Madrid, Spain; margrijo@ucm.es; 5Hormone Laboratory, Haukeland University Hospital, N-5021 Bergen, Norway; johan.ferno@uib.no; 6Institute for Endocrinology and Diabetes—Molecular Endocrinology, Center of Brain Behavior and Metabolism CBBM, University of Lübeck, 23562 Lübeck, Germany; jens.mittag@uni-luebeck.de

**Keywords:** thyroid hormones, AMPK, hypothalamus, brown adipose tissue, browning

## Abstract

Besides their direct effects on peripheral metabolic tissues, thyroid hormones (TH) act on the hypothalamus to modulate energy homeostasis. However, since most of the hypothalamic actions of TH have been addressed in studies with direct central administration, the estimation of the relative contribution of the central vs. peripheral effects in physiologic conditions of peripheral release (or administration) of TH remains unclear. In this study we used two different models of peripherally induced hyperthyroidism (i.e., T4 and T3 oral administration) to assess and compare the serum and hypothalamic TH status and relate them to the metabolic effects of the treatment. Peripheral TH treatment affected feeding behavior, overall growth, core body temperature, body composition, brown adipose tissue (BAT) morphology and uncoupling protein 1 (UCP1) levels and metabolic activity, white adipose tissue (WAT) browning and liver metabolism. This resulted in an increased overall uncoupling capacity and a shift of the lipid metabolism from WAT accumulation to BAT fueling. Both peripheral treatment protocols induced significant changes in TH concentrations within the hypothalamus, with T3 eliciting a downregulation of hypothalamic AMP-activated protein kinase (AMPK), supporting the existence of a central action of peripheral TH. Altogether, these data suggest that peripherally administered TH modulate energy balance by various mechanisms; they also provide a unifying vision of the centrally mediated and the direct local metabolic effect of TH in the context of hyperthyroidism.

## 1. Introduction

The relationship between thyroid hormones’ (TH; T3: triiodothyronine and T4: thyroxine) status and whole-body energy balance was the described as early as the end of the 19th century [1], with numerous clinical examples in patients with thyroid dysfunction [2,3]. Hyperthyroidism and/or thyrotoxicosis induce a hypermetabolic state with a raising basal metabolic rate (BMR) and feeding-independent weight loss [2,4,5,6,7]. In contrast, a reduction in TH levels is associated with reduced BMR and a tendency to weight gain despite reduced food intake [3,8]. Several of the molecular and cellular mechanisms underlying TH action in the metabolic organs have been identified. In the brown adipose tissue (BAT), TH increase the stimulatory action of noradrenaline (NE) and enhance the cAMP-mediated rise in uncoupling protein 1 (UCP1) mRNA expression [7,9,10,11,12]. In the liver, TH induce hepatic gluconeogenesis and glycogenolysis [13,14] and regulate the tightly interconnected lipid metabolism [15,16,17]. In addition, TH effects on core body temperature, body composition and white adipose tissue (WAT) browning have also been described [10,18,19,20,21,22]. Notably, evidence from the last fifteen years has demonstrated that, in addition to the direct peripheral action of TH, these hormones also act within the hypothalamus to regulate metabolic effects [11,18,19,23,24,25,26,27,28]. Most of this evidence is based on experimental rodent models of central hyperthyroidism (i.e., intracerebroventricular and/or nucleus-specific hypothalamic TH administration) [16,18,23,26,29], which allows for a detailed exploration of the hypothalamic TH-induced outputs to the peripheral targets. However, in the more physiological condition of peripherally released TH, these centrally driven effects interact and integrate with the direct tissue TH action [10], and one of the most challenging issues is to dissect the relative contribution of each component in different serum TH concentrations and physiological or pathological conditions. In fact, the amount of peripheral TH reaching and acting in the central nervous system (CNS) depends on the expression and activity of the blood–brain barrier (BBB) transporters and the subsequent neuronal, astrocyte and tanycytes deiodination and metabolism, each passage being differentially regulated depending on TH status (euthyroidism, hypo- or hyperthyroidism) and energy homeostasis (positive, negative or neutral energy balance) [30,31,32,33,34]. To address the relative contribution of the central vs. peripheral actions of TH, here we used a murine model of peripherally (oral) administration of T4 or T3 to induce hyperthyroidism and asses the relative contribution to the hypothalamic pool of TH, as well as their metabolic effects.

## 2. Materials and Methods

### 2.1. Animals and Housing Conditions

Adult male wild-type mice (C57BL/6 background; 8–10 weeks old, Centro de Biomedicina Experimental; Santiago de Compostela, University of Santiago de Compostela; Santiago de Compostela, Spain) were housed in groups (4 mice/cage) with an artificial 12-h light (8:00 to 20:00)/12-h dark cycle, under controlled temperature (22–23 °C) and humidity conditions. Before starting the experimental procedure, the animals underwent a 7-day period of acclimatation to the facility and to the handling procedure under non-stressful conditions. During this time, free access to standard laboratory chow diet (STD, SAFE A04: 3.1% fat, 59.9% carbohydrates, 16.1% proteins, 2.791 kcal/g; Scientific Animal Food & Engineering; Nantes, France) and tap water was available. All experiments and procedures were performed in agreement with International Law on Animal Experimentation and USC Ethical Committee (project ID 15010/14/006 and 15012/2020/10).

### 2.2. Induction of Hyperthyroidism and Monitoring

Mice were treated with TH in drinking water (Figure 1) at the following concentrations: 1 mg/L L-Thyroxine (T2501, Sigma-Aldrich; St. Louis, MO, USA) in 0.01% BSA (BSA, Albumin, from bovine serum, Sigma-Aldrich; St. Louis, MO, USA) or 0.5 mg/L 3,3′,5-Triiodo-L-thyronine (T32877, Sigma-Aldrich; St. Louis, MO, USA) in 0.01% BSA for 14 days [10]. Control mice received 0.01% BSA (vehicle treatment). The animals were kept housed in groups (*n* = 4 per cage) and were treated for a total of 14 days by ad libitum access to drinking water containing either T4, T3, or vehicle as previously detailed. Free access to standard chow diet was also provided for the duration of the treatment. The whole experiment was repeated 3 times. Body weight and food intake were measured daily at the same hour with a precision scale, and water intake was measured with a graduated cylinder. Body length, i.e., nose-to-tail distance, was measured with a ruler at the end of the treatment (Figure 1).

### 2.3. Core Body Temperature, BAT and Tail Temperature Measurements

Core body temperature was measured at the end of the treatment using a rectal probe connected to digital thermometer (BAT-12 Microprobe Thermometer; Physitemp; Clifton, NJ, USA). Skin temperature surrounding BAT, as well as the temperature at the base of the tail, were recorded at the end of the treatment with an infrared camera (B335: Compact Infrared Thermal-Imaging Camera; FLIR; West Malling, Kent, UK) and analyzed with a specific software package (FLIR Tools Software, FLIR; West Malling, Kent, UK), as previously shown [16,35,36,37,38,39,40,41,42,43,44,45]. For each image, the average temperatures were calculated as the average of 2 pictures/animal.

### 2.4. Sample Processing

Animals were sacrificed on day 14 (end of treatment) by cervical dislocation. From each animal, the whole hypothalamus (for RIA), or the ventromedial hypothalamus (dissected from the whole hypothalamus, for WB) were extracted, as well as the liver, the BAT, the gonadal/epididymal WAT (gWAT) and the subcutaneous/inguinal WAT (scWAT). The tissues were weighted freshly with a precision scale and subsequently harvested on ice. The blood of the animals was collected in 1.5 mL tubes and centrifuged during 15 min at 2000 rpm to separate the serum. Samples were stored at −80 °C until further processing.

### 2.5. Radioimmunoassay

Total T4 (tT4) and T3 (tT3) determination in serum and tissues was performed as previously described, with some minor changes [46,47]. High specific activity ^125^I-T3 and ^125^I-T4 (3000 μCi/μg) were labeled with ^125^I (NEZ033A; Perkin Elmer; Waltham, MA, USA using as substrates T2 (D0629; Sigma-Aldrich; St Louis, MO, USA) and T3 (T2877; Sigma-Aldrich; St Louis, MO, USA), respectively. The separation of the labeled products was performed by ascending paper chromatography for 16 h in the presence of butanol:ethanol:ammonia 0.5 N (5:1:2) as solvent. The ^125^I-T3 and ^125^I-T4 were eluted and kept in ethanol at 4 °C.

For hormonal determinations in serum, the blood of the animals was collected in specific tubes (BD Vacutainer; Plymouth, UK) and centrifuged during 15 min at 2000 rpm to obtain the serum fraction. T3 and T4 were extracted with methanol (1:6) from individual 80 μL aliquots of serum, evaporated to dryness and taken up in the RIA buffer; 0.04 M of phosphate buffer pH 8, with 0.2% bovine serum albumin (BSA) and 0.6 mM of merthiolate (T-5125; Sigma-Aldrich; St. Louis, MO, USA). For hormonal determination in the hypothalamus, the extraction of T4 and T3 was performed by homogenization of the frozen tissue in methanol, followed by a second extraction with chloroform–methanol (2:1) and purification using AG 1-X2 resin (140–1251; Bio-Rad Laboratories; Hercules, CA, USA) with 70% acetic acid as eluent. Samples were evaporated to dryness and resuspended in RIA buffer. To estimate the yield of the TH extraction, small amounts of ^125^I-T3 and ^125^I-T4 were added as internal tracers, both in serum and tissue initial homogenates. The detection ranges for RIA assay were 0.4–100 pg T3/tube and 2.5–320 pg T4/tube.

### 2.6. Quantification of Lipids

Tissues (30 mg) were homogenized with 10 volumes of ice-cold phosphate-buffered saline (PBS) in a Potter homogenizer (20 strokes). Fatty acids were measured in homogenates using a kit (Wako Chemicals; Richmond, VA, USA) and triglycerides (TG) were quantified as described [16]. Briefly, 100 mg of tissue were homogenized, lipids were extracted homogenates and TG were measured in the lipid extract with a kit (A. Menarini Diagnostics; Ripolli, Italy).

### 2.7. Real-Time Quantitative RT-PCR

Real-time PCR (TaqMan; Applied Biosystems; Foster City, CA, USA) was performed using specific primers and probes (Appendix A) as previously described [16,23,43,48,49]. Total RNA was isolated by using Trizol Reagent (Invitrogen; Carlsbad, CA, USA) according to the manufacturer’s protocol (RNA was precipitated with chloroform and isopropanol, washed with 75% ethanol, and finally dissolved in RNase-free water). cDNA synthesis was performed with M-MLV enzyme (Invitrogen; Carlsbad, CA, USA) following the supplier’s protocol. All reactions were carried out using the following cycling parameters: 50 °C for 2 min, 95 °C for 10 min followed by 40 cycles of 95 °C for 15 s, 60 °C for 1 min. Values were expressed in relation to hypoxanthine-guanine phosphoribosyl-transferase (Hprt) levels.

### 2.8. Histology and Immunohistochemistry

BAT and WAT samples were fixed in 10% formalin and maintained in 70% ethanol prior to paraffin embedding. Subsequently, they were included in paraffin, cut with a microtome, placed in slides, and stained with hematoxylin and eosin (H&E) or incubated with UCP1 antibody (Abcam, Cambridge, UK), followed by DAB staining (Envision, DAKO; Glostrup, Denmark), as previously shown [20,36,43,44,48,50]. Images were taken with a digital camera Olympus XC50 (Olympus Corporation; Tokyo, Japan) at 20×. Hepatic lipid content was analyzed by Red Oil O staining. Hepatic frozen sections were cut (8 μm), fixed in 10% buffered formaldehyde and subsequently stained in filtered Oil Red O (Sigma-Aldrich; St. Louis, MO, USA), washed in distilled H_2_O, counterstained with Harris hematoxylin (Bio-Optica; Milan, Italy) and washed distilled H_2_O. Sections were mounted in aqueous mounting medium (Bio-Optica; Milan, Italy). Images were taken with the digital camera Olympus XC50 (Olympus Corporation; Tokyo, Japan) at 40×, and quantified with ImageJ-1.33 (NIH; Bethesda, MD, USA).

### 2.9. Western Blotting

Protein lysates from ventromedial hypothalamus, liver and BAT were homogenized in lysis buffer (consisting of a mix of 0.05 M Tris-HCl, 0.01 M EGTA, 0.001 M EDTA, 0.016 M Triton X-100, 0.001 M sodium orthovanadate, 0.05 M sodium fluoride, 0.01 M sodium pyrophosphate and 0.25 M sucrose, made up with distilled water and adjusted to pH 7.5; all of them from Sigma-Aldrich; St. Louis, MO, USA) and freshly added protease inhibitor cocktail tablets (Roche Diagnostics; Indianapolis, IN, USA). The protein concentration was determined by the Bradford method (protein assay dye concentrate, Bio-Rad Laboratories; Hercules, CA, USA), and the total protein content of the tissues was calculated. The protein lysates were subjected to SDS-PAGE, electrotransferred to polyvinylidene difluoride membranes (PVDF; Millipore; Billerica, MA, USA) with a semidry blotter and probed with antibodies against acetyl-CoA carboxylase alpha (ACCα), AMPKα1, AMPKα2 (Millipore; Billerica, MA, USA), phospho-acetyl-CoA carboxylase alpha (pACCα (Ser79), phopho-AMPKα (pAMPKα) (Thr172), fatty acid synthase (FAS), UCP1, uncoupling protein 3 (UCP3) (Abcam; Cambridge, UK); PGC1α (Santa Cruz biotechnology); α-tubulin, β-actin or GADPH (Sigma-Aldrich; St. Louis, MO, USA), as previously described [10,16,18,23,35,36,37,43,44,48]. Each membrane was then incubated with the corresponding secondary antibody: anti-mouse or anti-rabbit (all of them from DAKO; Glostrup, Denmark). The membranes were exposed to an X-ray film (Fujifilm; Tokyo, Japan) and developed using developer (Developer G150; AGFA HealthCare; Mortsel, Belgium) and Fixator (Manual Fixing G354; AGFA HealthCare; Mortsel, Belgium). Autoradiographic films were scanned, and the band’s signal was quantified by densitometry using ImageJ-1.33 software (NIH; Bethesda, MD, USA). Values were expressed in relation to β-actin (ventromedial hypothalamus), α-tubulin (BAT) or GADPH (liver). Representative images for all proteins are shown; in all Figures showing images of gels, all the bands for each picture always come from the same gel, although they may be spliced for clarity.

### 2.10. Statistical Analysis

Statistical analysis was conducted using GraphPad Instat 3.10 and GraphPad Prism 8 Software (GraphPad Software; La Jolla, CA, USA). Data are expressed as MEAN ± SEM as percentage of the controls (vehicle-treated mice). Statistical significance was determined by ANOVA followed by post hoc two-tailed Tukey test. *p* < 0.05 was considered significant.

## 3. Results

### 3.1. Oral TH Administration Leads to a Significant Increase in Both Serum and Hypothalamic TH Levels

Treatment with T4 induced a significant raise in the circulating levels of both T4 (4.3-fold increase vs. vehicle-treated) and T3 (3-fold increase vs. vehicle-treated) while T3-treated mice displayed a 3-fold increase of T3 and a 75% decrease in T4, in line with the negative feedback loop of the hypothalamic–pituitary–thyroid (HPT) axis (Figure 2A–C). Therefore, our treatment achieved a clear hyperthyroid status. Then, we assessed the correspondent intra-hypothalamic concentration of T4 and T3 in each experimental group. In T4-treated mice, the hypothalamic levels of both THs were increased when compared with vehicle-treated mice (2.5-fold and 3.5-fold for T4 and T3, respectively) in line with the corresponding changes in their serum values. On the other hand, the hypothalamic TH levels under T3 treatment did not fully match the corresponding changes in serum. In fact, despite the 4.5-fold increase in hypothalamic T3, T4 levels were not significantly decreased, despite the consistent fall in serum T4 (Figure 2B–D). As an additional indicator of the treatment-induced changes in TH homeostasis, local metabolism and tissue action, we calculated and compared the T4/T3 ratio in serum and hypothalamus. This index resulted to be 70:1 in vehicle-treated mice, almost unchanged upon T4 treatment, and significantly decreased in T3-treated mice (5:1; *p* < 0.001 vs. vehicle) (Figure 2E). Conversely, the hypothalamic T4/T3 ratio was 5:1 in vehicle-treated mice and significantly decreased in both T4- (3:1, *p* = 0.011 vs. vehicle) and T3-treated mice (0.8:1, *p* < 0.0001 vs. vehicle) (Figure 2F).

### 3.2. Orally Induced Hyperthyroidism Leads to Changes in Energy Balance

TH treatment induced significant changes in body weight, body composition and food intake in both experimental groups. During the two weeks of treatment, we reported a progressive body weight gain in both T3- and T4-treated mice when compared with euthyroid (vehicle-treated) controls (Figure 3A). A gradual rise in daily food intake was also reported in both the T4- and T3-treated groups during the first week of treatment with respect to controls (data not shown) with a significant difference in the cumulative food intake throughout the experiment (Figure 3B). Significant differences were also found when comparing the body length (i.e., nose-to-base-of-tail distance) measured at the end of the experiment, which was consistently longer in hyperthyroid mice than in euthyroid mice (Figure 3C). The fat mass was also measured. At sacrifice, iBAT, gWAT and scWAT were extracted and weighted freshly and values were expressed as raw or normalized by body weight for inter-group comparison. The iBAT weight was significantly higher in both the hyperthyroid mouse groups than in the controls, and in T3-treated micewhen compared with T4-treated mice. T3-treated mice also displayed a significant reduction in the gWAT depots when compared to the vehicle-treated ones, but no difference between the groups was observed for the scWAT mass (Figure 3D,E).

Finally, at the end of the treatment, we analyzed the changes in temperature. While our data showed no differences in BAT temperature (Figure 3F,G), tail temperature, a well-known thermoregulatory mechanism in rodents modulated by TH [40] was increased upon treatment with T3, but not T4 (Figure 3H). Both TH increased core body temperature (Figure 3I).

### 3.3. Orally Induced Hyperthyroidism Impacts BAT Metabolism

The BAT mass was significantly bigger in hyperthyroid mice, with larger lipid droplets and adipocyte size, conferring a whitened appearance (Figure 4A,B). The increased lipid content was confirmed by TG measurement (Figure 4C) and was associated with elevated total protein content in the tissue (Figure 4D). Deiodinase 2 (DIO2) mRNA levels were downregulated in the hyperthyroid models, with a concomitant trend towards the upregulation of DIO3, representing a pattern that is consistent with the condition of TH excess. No significant changes in the expression of thyroid hormone receptors (TRs, TRα and TRβ) emerged. In addition, a downregulation of beta 3 adrenergic receptor (Adrb3) mRNA was reported in the T3-treated group, likely indicating a reduced BAT sensitivity to NE and therefore reduced sympathetic stimulation (Figure 4E). Western blot analyses demonstrated increased UCP3 protein levels in both T3 and T4 hyperthyroid animals, with UCP1 being only increased in the BAT of T3-administed mice; no changes in other thermogenic markers, such as PGC1α, were detected (Figure 4F,G).

### 3.4. Orally Induced Hyperthyroidism Promotes Browning of WAT

Upon H&E staining, both the gWAT and scWAT of hyperthyroid mice displayed a multilocular, brown-like pattern and a significant reduction in the adipocyte area (Figure 5A,B,E,F). Subsequent IHC analysis revealed a significant increase in UCP1 staining in both subtypes of white fat of the two treated groups (Figure 5C,G). This result was later confirmed by UCP1 mRNA expression analysis, leading to an equivalent result in the same tissues and groups of treatment (Figure 5D–H). Moreover, among the other tested thermogenic markers, DIO2 mRNA was also increased in both subtypes of WAT in T3-treated mice and in the scWAT of T4-treated mice as well (Figure 5D–H). In addition, scWAT, but not gWAT, displayed an enhanced expression of cell death activator CIDE-A (CIDEA) in both T3- and T4-treated animals and of PR/SET Domain 16 (PRDM16) and peroxisome proliferator-activated receptor gamma (PPARγ) in the T3-treated group (Figure 5D–H).

### 3.5. Orally Induced Hyperthyroidism Promotes Hepatic Lipid Accretion

Both hepatic Oil Red O staining and TG content were increased in hyperthyroid mice, indicating lipid accumulation in the liver, which reached significant differences in T3-treated mice (Figure 6A–C). Protein expression analysis revealed an increased expression of FAS in the T3-treated group, consistent with enhanced de novo lipogenesis, whereas the levels of the pAMPKα and pACCα did not differ among the three groups (Figure 6D,E). Both total ACCα and AMPKα1 subunit protein levels were found to be increased in the T3-treated group (Figure 6D,E). 

### 3.6. Orally Induced Hyperthyroidism Inhibits AMPK Signaling in the Ventromedial Hypothalamus

Following the demonstration that our employed oral treatment significantly raised the hypothalamic levels of both T4 and T3, we tested whether those TH concentrations were able to trigger the abovementioned pathways. T3-treated hyperthyroid mice showed a marked decrease in hypothalamic pAMPKα and the AMPKα1 and α2 subunits, as well as pACCα. T4-treated mice displayed a significant reduction in pAMPKα and AMPKα1 and a non-significant trend for pACCα (Figure 7A,B). Overall, this evidence suggests that the peripheral administration of TH, modulating BAT thermogenic program and WAT, is associated with changes in AMPK signaling at the hypothalamic level.

## 4. Discussion

The discovery of the central TH action on body metabolism challenged the classical view of the peripheral effects of these hormones. However, despite this evidence, the exact contribution of the central effect to each of the metabolic actions of TH remains unclear. Some evidence has been obtained by genetic manipulation; for example, it is known that in the context of systemic hyperthyroidism, the knockdown of TRs in the VMH reversed the effects of TH on BAT thermogenesis [23]. Recent evidence has also been obtained from mice lacking both monocarboxylate transporter 8 (MCT8) and organic anion transporting polypeptide 1C1 (OATP1C1). This M/O double knock-out (exhibiting elevated T3 levels, while their brain shows a marked hypothyroidism) lacks the pyrexic response to the peripheral hyperthyroidism, suggesting that the central actions of TH are required for this hyperthyroid phenotype [25]. Nevertheless, most of the central effects have been studied in models of direct hypothalamic TH treatment, and therefore, one of the most challenging issues is the estimation of the relative contribution of the central vs. peripheral component, and to document to what extent the oral ingestion of T4 and T3 may lead to different effects on the different components of energy balance [10].

Our data showed that upon oral treatment, T4 and T3 levels significantly increased both in the serum and in the hypothalamus of the corresponding treatment groups, confirming the peripheral and central hyperthyroid status. The T3 concentrations also increased in both the serum and hypothalamus of the T4-treated mice, which is consistent with effective BBB crossing of the exogenous hormones and proper cellular deiodination. Interestingly, in the T3-treated group, the deep fall in serum T4 levels was not accompanied by a contemporary decrease in the T4 hypothalamic concentration, which instead remained unchanged relative to controls, even in the presence of much higher levels of hypothalamic T3. The molecular basis, as well as the physiological significance of this T4 preservation in the hypothalamus, remains to be clarified.

Human hyperthyroidism has classically been associated with body weight loss despite hyperphagia due to excess energy waste [4,5]. In mice, the overall effect on body weight is more variable, due to the different thermal physiology and the lifelong body growth pattern [10,51]. In the present study, mice treated with peripheral TH expectedly increased their food intake but did not experience body weight loss. Instead, they displayed an early and progressive weight gain during the two weeks of treatment, which was significantly more pronounced than that observed for the euthyroid control group. The increase in body weight did not seem to depend on fat mass deposition, as there were no changes in scWAT depots or a reduction in the gWAT depots but it was instead accompanied by overall body growth (i.e., increased body length). The same effect on body weight and composition, plus an increased lean mass, has recently been reported in other studies [10,21] and is in line with the fact that TH promote linear body growth in rodents, species which keep the capacity of growing during all their life [10,51,52,53,54]. In keeping with this, it has been demonstrated that the orexigenic effect of TH does not only occur as compensatory reaction in response to energy wasting but also represents an independent early event and can even be induced by low doses of T3, which do not affect energy expenditure [18,55]. Moreover, WAT depots were not only reduced in size, such as those observed for gWAT, but also acquired the features of browning, a phenomenon which had already been described both in the peripherally and centrally induced models of hyperthyroidism [10,20]. On the other hand, the iBAT depots increased in size with larger lipid droplets despite reduced expression of local lipogenic enzymes, suggesting enhanced peripheral uptake, which is in line with what has previously been observed after central T3 administration [16]. Notably, even if this could resemble the obesity-related BAT whitening process [56], there is no evidence of the whitening-associated cellular dysfunction. Instead, an increase in the total protein content emerged, including elevated UCP1 and UCP3 protein expression, which is consistent with an effective BAT recruitment, enhanced uncoupling capacity and function [21]. To better investigate this issue, the determination of the total UCP1 content in the whole BAT depot would allow us to better define the tissue recruitment status [19,57]. An increased lipid accumulation in the liver was also observed after thyroid hormone treatment. In contrast to the observation in BAT, this may be explained by enhanced hepatic de novo lipogenesis. Interestingly, this process can be induced by direct TR activation in hepatocytes [58] but also by targeting AMPK and c-Jun N-terminal kinases (JNK1) activity in the steroidogenic factor 1 (SF1) neurons of the VMH [16]; the relative contribution of each pathway will be the subject of further studies.

It has been demonstrated that the central modulation of metabolism by TH depends on targeting AMPK in specific groups of hypothalamic neurons [10,16,23,29,59,60]. Then, to assess if the reported hypothalamic TH concentrations were high enough to trigger those pathways, we determined the state of the AMPK downstream factors in the ventromedial hypothalamus. In the T4- and T3-treated groups, the AMPK phosphorylation was inhibited, as has been shown upon direct central T3 administration [16,18,23] and is in line with previous reports on hyperthyroid rats [18,23]. Interestingly, we observed a milder effect on the hypothalamic AMPK downstream in the T4 group despite similar local T3 levels. We hypothesized that this could depend on (i) the differences in the activity of “pure” T3 vs. the one derived by peripheral (serum and/or tissue-specific) deiodination; (ii) a possible independent role of intact T4 itself, as, despite the similar T3 hypothalamic content, the T4/T3 ratio substantially differs in the two treatment groups.

Overall, most of our results recapitulated the work of Johann et al. [10], confirming that this protocol of TH administration in ad libitum drinking water is a suitable model for the study of peripherally induced mice hyperthyroidism. However, there were a few exceptions. For example, our T4-treated mice were fully hyperthyroid, while the T4 model of Johann et al. was milder, with no significant elevation in total serum T3 [10]. This inconsistency could depend on the variation in the animal husbandry between the two facilities or on the substrain used (wild-type C57/BL6NCr vs. C57BL/6 background). Another discrepancy is represented by the body temperature in the T4-treated mice, which was elevated in our work and almost unchanged in the previous study, possibly because of the different grades of hyperthyroidism, housing conditions and/or the abovementioned metabolic differences between the substrains. In relation to the hepatic AMPK signaling, some subtle differences were observed. In rats, T4 injections promoted decreased hepatic pAMPK (and pACC) and increased FAS [16,23]; on the other hand, after oral T3 and T4 in mice no changes were found in pAMPK or pACC, still in the context of increased FAS. This apparent lack of effects in the mouse setting does not account for a differential physiological effect: increased de novo lipogenesis (because of FAS increased expression) and lipid accretion, in both rats [16,23] and mice. The reasons for the lack of effect on AMPK in the mouse setting are unclear, but they could be related to the response to parasympathetic stimuli [16], which may vary according to the body energy requirements. Furthermore, at the mechanistic level, the possible interplay of TH and AMPK with the mTOR signaling in the liver may also play a role [9,10,11]. Finally, the hypothalamic AMPK inhibition was only observed in our current study but not in the previous one [10]; however, the hypothalamic T3 levels in that model were only around 3-fold elevated, while the T3 treatment in the present study resulted in an around 5-fold elevation. This corroborates the hypothesis that the central effect of TH on hypothalamic AMPK is strongly dose-dependent, as suggested recently [12], and can be triggered if hypothalamic T3 levels exceed a defined threshold, which might be set by the compensatory DIO3 increase in neurons that usually protects these cells against hyperthyroidism [61].

Our data provide interesting insights into the current debate on the thyroid thermogenesis, for which the molecular details remain to be clarified [6,62,63]. In this regard, we found in our model of peripheral hyperthyroidism: (i) a significant increase in core body temperature under both T4 and T3 treatment; (ii) a significant raise in the tail temperature in the T3-treated group, indicating a co-existence of heat production and dissipation [40] and (iii) no changes in the temperature of the interscapular skin above BAT, suggesting that the origin of the heat excess does not depend on an increased brown fat thermogenesis. The observation that the increased BAT uncoupling capacity is not translated in increased BAT temperature seems to contrast with the clearly demonstrated thermogenic action of central T3 [16,20,23,25]. However, it was also previously shown that peripherally induced thyroid thermogenesis is a UCP1-independent process [21], and that both BAT recruitment and WAT browning give only minor contributions to the core temperature elevation of the chronic peripheral hyperthyroid state [10,21]. This may be explained by different phenomena. First, despite an increased amount of circulating TH, the local decrease in DIO2 expression may reduce the BAT intracellular T3 production and activity. Another role may be played by the TH-dependent modulation of BAT sensitivity to NE, involving a reduced expression of β-adrenergic receptors in BAT that prevents the hypothalamic signals to act [10,12], overall making the brown fat less active [10]. In line with this, we also found the expression of Adrb3 to be reduced in BAT in peripherally T3-treated animals, further supporting the existence of a local physiological feedback mechanism that uncouples BAT thermogenesis from central stimulation. This may constitute an additional defense to TH-induced overheating, which would usually be observed from increased sympathetic tone after ventromedial AMPK inhibition [16,19,23,29].

An important issue that would require clarification is the timing of the central vs. peripheral actions of T3. Elucidating this issue would address whether the increased uncoupling capacity is: (i) an early event elicited by hypothalamic T3 [16,19,23,29] and later abrogated by the peripheral action of TH on β3-AR expression or (ii) a peripheral direct action occurring after blockage of the central action. In our view, the first option would be more likely; the central effect would induce an acute metabolic response, including enhanced BAT lipolysis and increased thermogenesis, with the net effect of negative energy balance and body weight loss [16,23,29,59]. On the other hand, the chronic peripheral TH administration gradually mediates a machinery of progressive metabolic changes, finally resulting in an increased overall uncoupling capacity (i.e., BAT recruitment and WAT browning) and a shift of the lipid metabolism from WAT accumulation to BAT and liver deposition, possibly to provide adequate fueling to face the increased metabolic activity. Notably, these two phenotypes (central/acute and peripheral/prolonged TH treatment) mimic the acute cold response and chronic cold adaptation, respectively, of both physiological responses regulated by TH [14,29,63]. To address this issue, a deeper investigation of the central vs. peripheral TH homeostasis in different thermal stress conditions in a dose- and time-dependent manner would be needed. This is an important issue in light of the ongoing search for a personalized replacement therapy in hypothyroidism, including the use of physiological dosing of a T4 and T3 combination, which in some studies appears to provide a better outcome in terms of body weight than T4 alone [64]. Additionally, recent reports showing that thyroid hormones are predictors of weight loss [65] underscores the need to uncover the relative contributions of both T4 and T3 in the different biological endpoints.

Altogether, these data represent a first step towards a better understanding of the interaction between the central and peripheral metabolic TH effects. Further work analyzing the nucleus-specific concentration of T3 and T4 in the hypothalamus, as well as the expression of TRs and TH transporters, will be needed to fully understand the central vs. peripheral actions of thyroid hormones and for the design of more rationale strategies against thyroid disorders and obesity.

## Figures and Tables

**Figure 1 nutrients-13-04204-f001:**
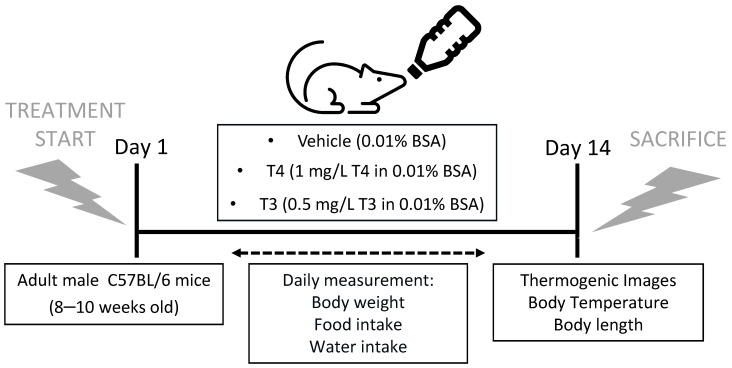
Experimental protocol and treatments. Study design showing the mice characteristics, treatments and analyses performed.

**Figure 2 nutrients-13-04204-f002:**
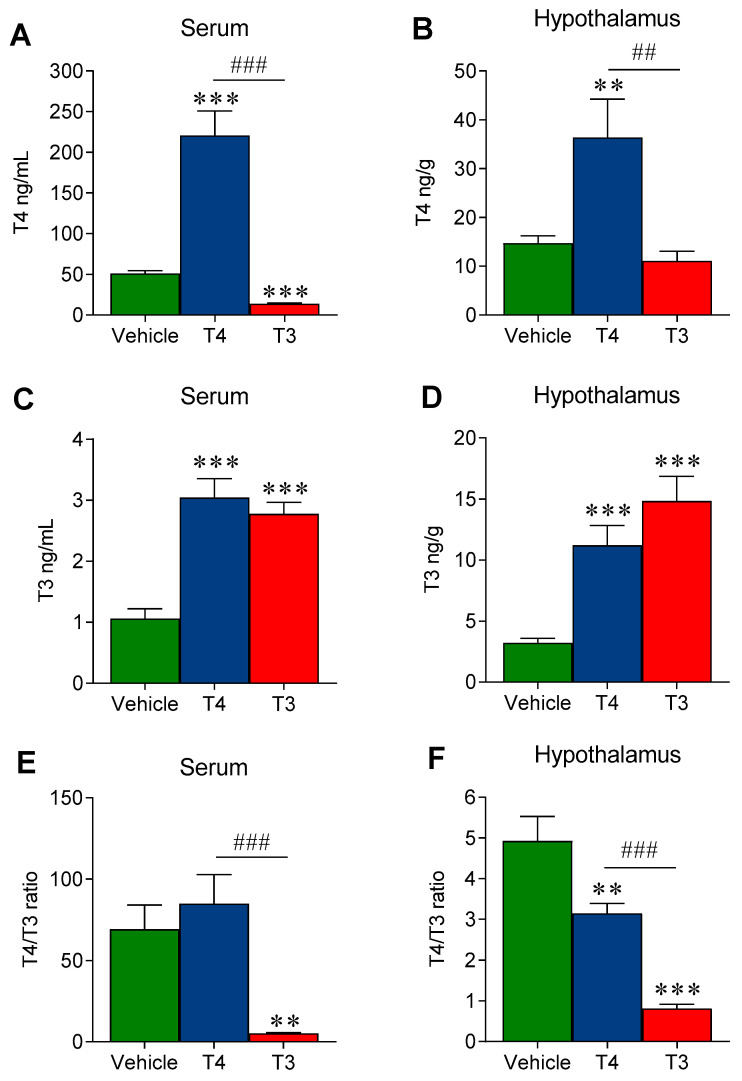
Treatment-specific changes in the serum and hypothalamic levels of total T4 and T3, and T4/T3 ratio:(**A**) Serum T4 levels; (**B**) Hypothalamic T4 levels; (**C**) Serum T3 levels; (**D**) Hypothalamic T3 levels; (**E**) Serum T4/T3 ratio; (**F**) Hypothalamic T4/T3 ratio. Values represented ad MEAN ± SEM. *n* = 8–12 mice/group. ** *p* < 0.01 T3 vehicle *** *p* < 0.001 T3 vehicle; ## *p* < 0.01 T4 vs. T3; ### *p* < 0.001 T4 vs. T3.

**Figure 3 nutrients-13-04204-f003:**
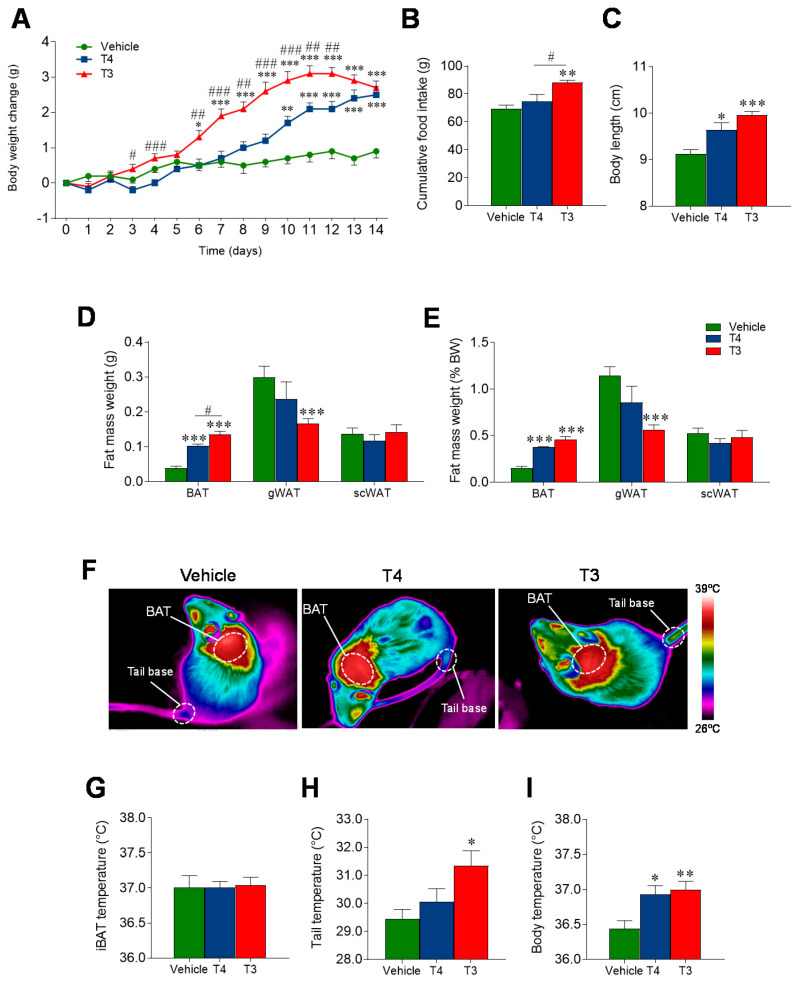
Effects of TH treatment on body mass, body composition, and temperature (**A**) Body weight change, (**B**) cumulative food intake and (**C**) end-of treatment body length (nose-to-tail distance) in the three experimental groups. (**D**–**E**) BAT, gWAT and scWAT weights measured freshly at dissection (**D**) and normalized for body weight (**E**). (**F**) Representative infrared pictures of iBAT and tail base of vehicle, T4, and T3 treated mice. Quantification of the (**G**) iBAT and (**H**) tail temperatures from representative infrared images. (**I**) Core body temperature measured at the end of treatment with a rectal probe. Values represented ad MEAN ± SEM. *n* = 8–12 mice/group. * *p* < 0.05 vs. vehicle; ** *p* < 0.01 T3 vehicle *** *p* < 0.001 T3 vehicle; # *p* < 0.05 T4 vs. T3; ## *p* < 0.01 T4 vs. T3; ### *p* < 0.001 T4 vs. T3.

**Figure 4 nutrients-13-04204-f004:**
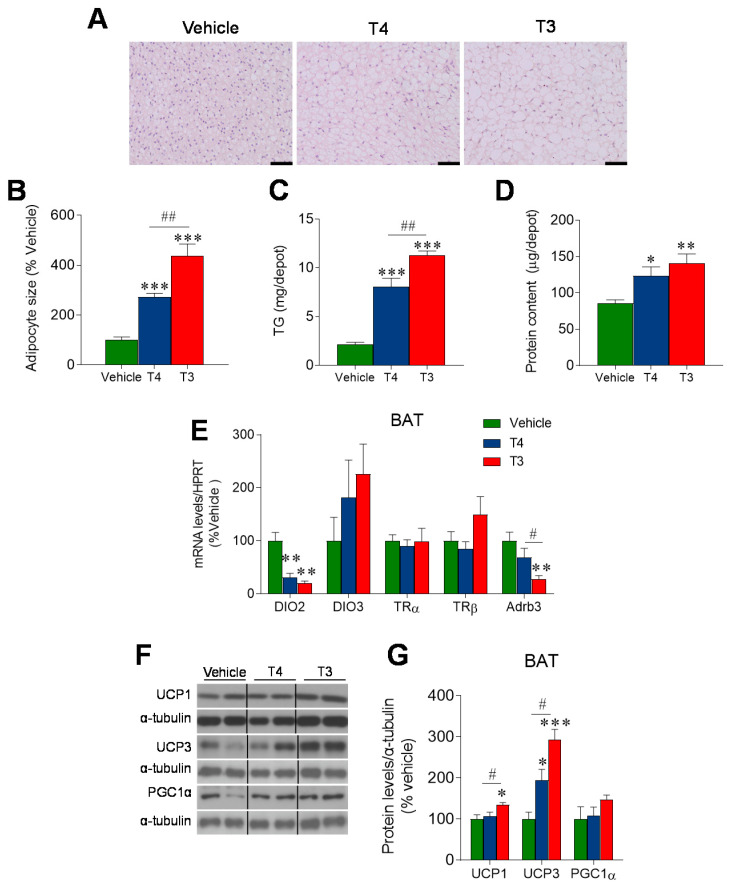
Effects of TH treatment on BAT: (**A**) Representative H&E staining of the iBAT (scale bar 50 μm) and (**B**) calculated average BAT adipocytes area in the three treatment groups, expressed as % of controls. (**C**) BAT lipid and (**D**) protein content. All data in MEAN ± SEM. *n* = 8–12 mice/group. (**E**) mRNA expression analysis of DIO enzymes, TRs, and Adrb3 in BAT. Data expressed as target mRNA/HPRT mRNA, % of vehicle and represented as MEAN ± SEM. *n* = 7–8 mice/group. (**F**,**G**) Protein levels of thermogenic markers in BAT. Data expressed as target protein/loading control, % of vehicle and represented as MEAN ± SEM. *n* = 4–5 (UCP3, PGC1α, AMPKα1, AMPKα2 and ACCα) or 8–10 (UCP1, pAMPKα, pACCα and FAS) mice/group. All the bands for each picture come always from the same gel, although they may be spliced for clarity. * *p* < 0.05 vs. vehicle; ** *p* < 0.01 T3 vehicle *** *p* < 0.001 T3 vehicle; # *p* < 0.05 T4 vs. T3; ## *p* < 0.01 T4 vs. T3.

**Figure 5 nutrients-13-04204-f005:**
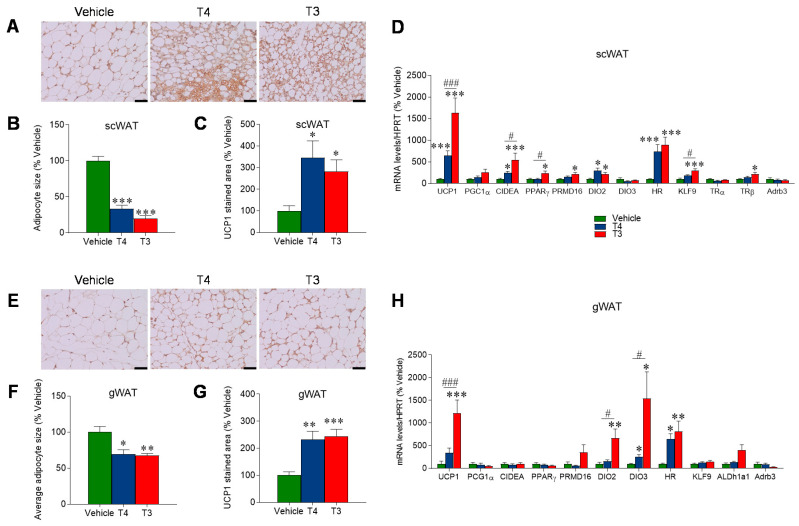
Effects of TH treatment on WAT:(**A**) Representative UCP-1 stainings of the scWAT in the three treatment groups (Scale bar 50 μm). (**B**) Calculated average scWAT adipocytes area and (**C**) UCP1 staining analysis, expressed as % of controls, in MEAN ± SEM. *n* = 8–12 mice/group. (**D**) mRNA expression analysis of browning markers, TH-induced genes, and TRs in scWAT of the three treatment groups. Data expressed as target mRNA/HPRT mRNA, % of vehicle and represented as MEAN ± SEM. *n* = 7–8 mice/group. (**E**) Representative UCP-1 stainings of the gWAT in the three treatment groups (Scale bar 50 μm). (**F**) Calculated average gWAT adipocytes area and (**G**) UCP1 staining analysis, expressed as % of controls, in MEAN ± SEM. *n* = 8–12 mice/group. (**H**) mRNA expression analysis of browning markers and TH-induced genes in gWAT of the three treatment groups. Data expressed as target mRNA/HPRT mRNA, % of vehicle, and represented as MEAN ± SEM. *n* = 7–8 mice/group. * *p* < 0.05 vs. vehicle; ** *p* < 0.01 T3 vehicle *** *p* < 0.001 T3 vehicle; # *p* < 0.05 T4 vs. T3; ### *p* < 0.001 T4 vs. T3.

**Figure 6 nutrients-13-04204-f006:**
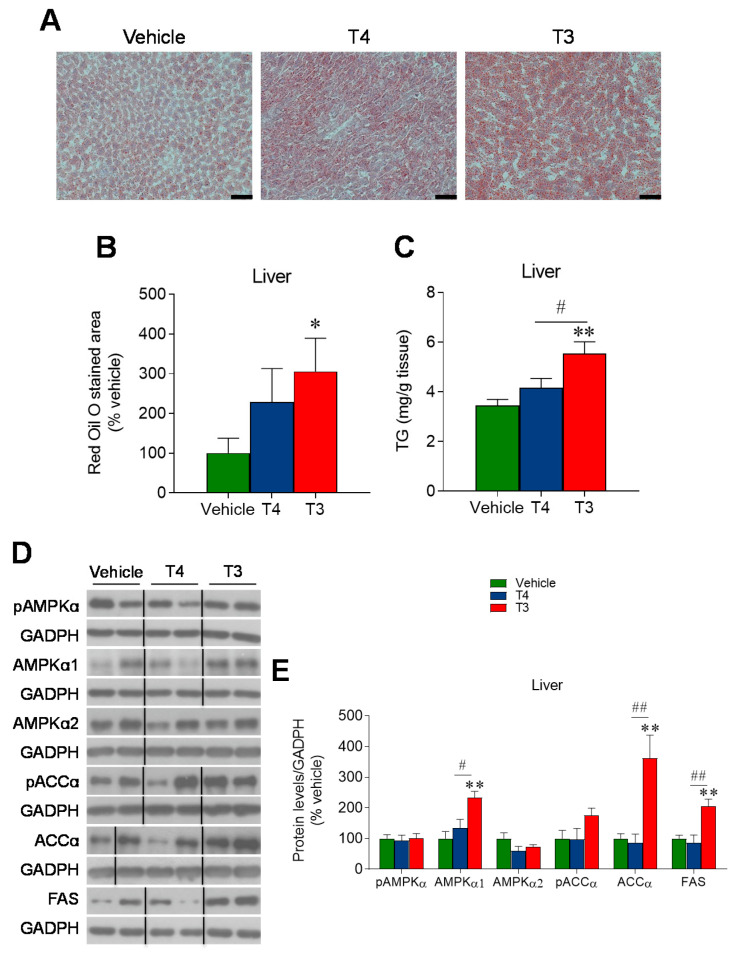
Effects of TH treatment on liver metabolism: (**A**) Representative liver Oil Red O staining (Scale bar 50 μm), (**B**) Oil Red O staining analysis and (**C**) liver TG content of the three treatment groups. *n* = 8–12 mice/group. Data expressed in MEAN ± SEM. (**D**,**E**) Protein levels of the AMPK pathway in the liver. Data expressed as target protein/loading control, % of vehicle and represented as MEAN ± SEM. *n* = 4–5 (AMPKα2) or 8 (pAMPKα, AMPKα1, pACCα and FAS) mice/group. All the bands for each picture always come from the same gel, although they may be spliced for clarity. * *p* < 0.05 vs. vehicle; ** *p* < 0.01 T3 vehicle; # *p* < 0.05 T4 vs. T3; ## *p* < 0.01 T4 vs. T3.

**Figure 7 nutrients-13-04204-f007:**
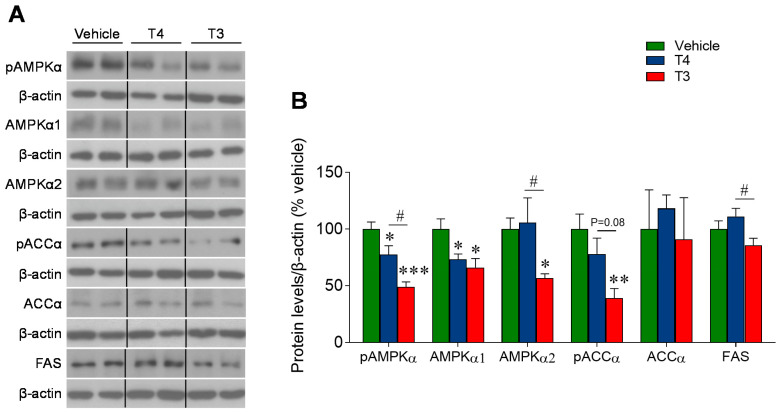
Effects of TH treatment on the AMPK pathway in the ventromedial hypothalamus: (**A**,**B**) Protein levels of the AMPK pathway in the ventromedial hypothalamus. Data expressed as target protein/loading control, % of vehicle and represented as MEAN ± SEM. *n* = 4–5 (AMPKα2, ACCα) or 8 (pAMPKα, AMPKα1, pACCα and FAS) mice/group. All the bands for each picture come always from the same gel, although they may be spliced for clarity. * *p* < 0.05 vs. vehicle; ** *p* < 0.01 T3 vehicle *** *p* < 0.001 T3 vehicle; # *p* < 0.05 T4 vs. T3.

## Data Availability

The data presented in this study are available on request from the corresponding author.

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
