# Peer review of "Orally Induced Hyperthyroidism Regulates Hypothalamic AMP-Activated Protein Kinase"

_nutrients, 2021, doi:10.3390/nu13124204_

Round 1
Reviewer 1 Report
Capelli et al. provides a manuscript reporting a study on the impact of orally-induced hyperthyroidism on adiposity and several parameters related to liver and hypothalamic function. The study concerns an overall subject (effects of thyroid hormones in peripheral tissues versus central effects; impact of thyroid hormone status on adiposity,..) largely studied previously but the specific approach followed in the current manuscript has a substantial extent of originality . Experimentation is appropriate although there are some aspects that would require clarification and further interpretation.
Specific points_
1) Whereas the effects reported in relation to WAT browning are very convincing, the interpretation of BAT data should be re-considered or at least discussed because this reviewers has the impression that there are actual signs of inhibition of activity in the hyperthyroid models. Considering that BAT temperature is unchanged but tail and core temperature are increased, wouldn't it mean that the relative "BAT-to-no BAT sites" in mice is in fact decreased? Concerning biomarkers, the calculations of UCP1 protein "per depot" instead of the relative UCP1 protein levels "per tubulin" would be more indicative of the actual thermogenic activity/capacity of BAT in such a model in which trophic changes in BAT may occur (see doi: 10.1016/j.bbalip.2013.01.009 for this rationale). Perhaps, expanding gene expression data to expanded markers of BAT thermogenic activity (as for WAT data) may help to clarify the actual status of BAT thermogenesis in the experimental models studied. In any case, reconsidering the overall discussion on BAT may be helpful for a balanced view.
2) The presentation of the size of adipose depots "normalized"per body weight is useful but perhaps insufficient has it is a calculation in which fat weight is corrected by a parameter (body weight) that also includes itself the weight of fat. The crude data of size of adipose depots without correction and the data corrected by size (e.g. length) not weight, parameters would be worth to be at least mentioned and discussed to strengthen the presentation of data.
3) Regarding BAT activity, perhaps a discussion similar to that provided in relation to central effects on the possibility that T3 from systemic circulation would act differently from intracellularly originated T3 after deiodination of T4 may be worth. Notice that the intense down-regulation of Dio2 would mean that local T3 generation is expected to be repressed.
4) The effects of hyperthyroidism on animal length are somewhat puzzling. May it be related to the relatively young animals used (8 week-old) despite being considered adults?
5) The description of adipose depots anatomical identity should be stated more explicitly in the Methods. As mice used are males, gonadal are expected to be epididymal, which should be stated. For subcutaneous, the WAT depot used as representative (inguinal?) should be stated at least in the Methods section.
6) There are a few typos only, but pelase do a round of revision in this regard (e.g. line 207, treated instead of "trated",...)
Author Response
REVIEWER#1 Overall comment: Capelli et al. provide a manuscript reporting a study on the impact of orally-induced hyperthyroidism on adiposity and several parameters related to liver and hypothalamic function. The study concerns an overall subject (effects of thyroid hormones in peripheral tissues versus central effects; impact of thyroid hormone status on adiposity,..) largely studied previously but the specific approach followed in the current manuscript has a substantial extent of originality. Experimentation is appropriate although there are some aspects
that would require clarification and further interpretation.
Response: We thank the Reviewer for the positive view of our manuscript and the excellent comments and suggestions. A detailed point-by-point response to the comments is included below.
Reviewer’s comment 1: Whereas the effects reported in relation to WAT browning are very convincing, the
interpretation of BAT data should be re-considered or at least discussed because this reviewer has the impression
that there are actual signs of inhibition of activity in the hyperthyroid models. Considering that BAT temperature is
unchanged, but tail and core temperature are increased, wouldn't it mean that the relative "BAT-to-no BAT sites"
in mice is in fact decreased? Concerning biomarkers, the calculations of UCP1 protein "per depot" instead of the
relative UCP1 protein levels "per tubulin" would be more indicative of the actual thermogenic activity/capacity of
BAT in such a model in which trophic changes in BAT may occur (see doi: 10.1016/j.bbalip.2013.01.009 for this rationale). Perhaps, expanding gene expression data to expanded markers of BAT thermogenic activity (as for
WAT data) may help to clarify the actual status of BAT thermogenesis in the experimental models studied. In any case, reconsidering the overall discussion on BAT may be helpful for a balanced view.
Response: We thank the Reviewer for these interesting points.
Reviewer’s comment 2: The presentation of the size of adipose depots "normalized" per body weight is useful
but perhaps insufficient has it is a calculation in which fat weight is corrected by a parameter (body weight) that
also includes itself the weight of fat. The crude data of size of adipose depots without correction and the data
corrected by size (e.g. length) not weight, parameters would be worth to be at least mentioned and discussed to
strengthen the presentation of data.
Response: We thank the Reviewer for his/her suggestion. Following recommendation, we show below the crude data of the weight of each fat depot (Rebuttal Figure 2). As the Reviewer can check the conclusions are the same to those obtained in the normalization by the body weight. These data are present in Figure 3D of the new version of the manuscript.
Reviewer’s comment 3: Regarding BAT activity, perhaps a discussion similar to that provided in relation to central
effects on the possibility that T3 from systemic circulation would act differently from intracellularly originated T3
after deiodination of T4 may be worth. Notice that the intense down-regulation of Dio2 would mean that local T3
generation is expected to be repressed.
Response: We thank the Reviewer for this interesting suggestion. Following this comment, we have discussed that intracellularly T3 production in BAT was likely reduced because of (i) decreased DIO2 expression and (ii) downregulated Adrb3 (Figure 4E), which is crucial for DIO2 induction by SNS; this again supports the idea that BAT capacity is up, but activation is down. These points have been discussed in the new version of the manuscript.
Reviewer’s comment 4: The effects of hyperthyroidism on animal length are somewhat puzzling. May it be related to the relatively young animals used (8 week-old) despite being considered adults?
Response: We apologize for not clarifying this better. TH promote linear body growth. In the case of rodents (rats and mice) growing does not stop after puberty; these species keep growing during all their life [3, 17-19]. As the Animal House Staff use to say over here: “if you see a big rat/mouse, it is likely an old rat/mouse”. We have reported similar growth in older mice (12-24 weeks) also treated with T3 or T4 [3]. However, it is true the growth rate is changing in response to developmental influences which are age-, sex- and body weight, among other factors. Because it is almost impossible to address all the developmental influences in our experimental model, we prefer to refrain of any speculation in this complex issue.
Reviewer’s comment 5: The description of adipose depots anatomical identity should be stated more explicitly in the Methods. As mice used are males, gonadal are expected to be epididymal, which should be stated. For
subcutaneous, the WAT depot used as representative (inguinal?) should be stated at least in the Methods section.
Response: The Reviewer is totally right. We apologize for the lack of clarity in these points.
As the Reviewer says, the gonadal depot was epididymal fat and the subcutaneous depot was inguinal. This information has been included in the new version of the manuscript.
Reviewer’s comment 6: There are a few typos only, but please do a round of revision in this regard (e.g. line 207, treated instead of "trated",...)
Response: We apologize for the mistakes; the new version of the manuscript has been proofread and the typos corrected.
We thank again the Reviewer#1 for the excellent comments and insight on our manuscript.
Reviewer 2 Report
This is an interesting study where researchers stimulated hyperthyroidism peripherally instead of centrally and observed the results. Although interesting, I was intrigued by the fact that weight gain occurred despite BAT stimulation and beiging. It would be good if researchers could comment on that. Also, show the entire gels with molecular weights of the WB´s would be helpful.
Author Response
REVIEWER#2 Overall comment: This is an interesting study where researchers stimulated hyperthyroidism peripherally instead of centrally and observed the results.
We thank the Reviewer for the positive view of our manuscript and the excellent comments and suggestions. A detailed point-by-point response to the comments is included below.
Reviewer’s comment 1: Although interesting, I was intrigued by the fact that weight gain occurred despite BAT
stimulation and beiging. It would be good if researchers could comment on that.
Response: We thank the Reviewer for his/her suggestion. The weight gain was mainly due to the linear growth of the animals, an effect that has been reported before [3, 17, 18, 20] and is among some of the most important actions of thyroid hormones [20]. Therefore, despite reduced adiposity (Figure 3D-E), the increased size of the animals (Figure 3C) accounts for higher body weight (Figure 3A).
Reviewer’s comment 2: Also, show the entire gels with molecular weights of the WB ́s would be helpful.
Response: We thank the Reviewer for his/her suggestion. All the original gels have been submitted, showing the chosen bands used in the panels. Those files were already uploaded in the first submission.
We thank again the Reviewer#2 for the excellent comments and insight on our manuscript.